# Real-World Colonoscopy Video Integration to Improve Artificial Intelligence Polyp Detection Performance and Reduce Manual Annotation Labor

**DOI:** 10.3390/diagnostics15070901

**Published:** 2025-04-01

**Authors:** Yuna Kim, Ji-Soo Keum, Jie-Hyun Kim, Jaeyoung Chun, Sang-Il Oh, Kyung-Nam Kim, Young-Hoon Yoon, Hyojin Park

**Affiliations:** 1Department of Internal Medicine, Division of Gastroenterology, Gangnam Severance Hospital, Yonsei University College of Medicine, Seoul 06273, Republic of Korea; sadts@yuhs.ac (Y.K.);; 2Waycen Inc., Seoul 06167, Republic of Korea; jisoo.keum@waycen.com (J.-S.K.);

**Keywords:** artificial intelligence, colon polyp, colon cancer, colonoscopy

## Abstract

**Background/Objectives**: Artificial intelligence (AI) integration in colon polyp detection often exhibits high sensitivity but notably low specificity in real-world settings, primarily due to reliance on publicly available datasets alone. To address this limitation, we proposed a semi-automatic annotation method using real colonoscopy videos to enhance AI model performance and reduce manual labeling labor. **Methods**: An integrated AI model was trained and validated on 86,258 training images and 17,616 validation images. Model 1 utilized only publicly available datasets, while Model 2 additionally incorporated images obtained from real colonoscopy videos of patients through a semi-automatic annotation process, significantly reducing the labeling burden on expert endoscopists. **Results**: The integrated AI model (Model 2) significantly outperformed the public-dataset-only model (Model 1). At epoch 35, Model 2 achieved a sensitivity of 90.6%, a specificity of 96.0%, an overall accuracy of 94.5%, and an F1 score of 89.9%. All polyps in the test videos were successfully detected, demonstrating considerable enhancement in detection performance compared to the public-dataset-only model. **Conclusions**: Integrating real-world colonoscopy video data using semi-automatic annotation markedly improved diagnostic accuracy while potentially reducing the need for extensive manual annotation typically performed by expert endoscopists. However, the findings need validation through multicenter external datasets to ensure generalizability.

## 1. Introduction

Artificial intelligence (AI) integration into medical imaging, particularly colonoscopy image analysis, has become increasingly relevant in clinical practice due to its potential to enhance polyp detection and diagnosis [1,2,3]. AI models supporting colonoscopy procedures generally fall into two categories: computer-aided detection (CADe), which focuses on detecting abnormalities, such as polyps, and computer-aided diagnosis (CADx), which classifies detected lesions based on their characteristics [4].

Despite advancements in AI models for colonoscopy, the lack of easily accessible, high-quality data for training and validating AI models remains a challenge [5]. Models trained exclusively on publicly available datasets frequently encounter difficulties due to discrepancies in image quality, diversity, and differences in clinical equipment, resulting in high sensitivity but notably low specificity in clinical settings [6,7].

This study aimed to address these limitations through a novel semi-automatic annotation method utilizing real patient colonoscopy videos from Gangnam Severance Hospital. Unlike previous methods relying heavily on manual annotation by expert endoscopists, our approach leveraged preliminary AI inference results for efficient selection of clinically relevant video frames, substantially reducing annotation effort. Although semi-automatic annotation is a common method in general computer vision tasks, our specific approach—AI-driven preliminary selection from colonoscopy video data—is designed explicitly for clinical practice integration.

This study demonstrates the feasibility of developing an accurate and efficient AI polyp detection model by integrating real-world clinical data, significantly reducing manual annotation burdens. Nevertheless, due to the single-center nature of the data collection, additional multicenter validation is necessary to ensure broader applicability.

## 2. Materials and Methods

### 2.1. Patients

Colonoscopy video data were retrospectively collected from patients who underwent routine colonoscopies at Gangnam Severance Hospital, a tertiary academic center, between April 2021 and April 2022. Inclusion criteria consisted of adults (≥18 years old) undergoing colonoscopy for indications including colorectal cancer screening, surveillance, or evaluation of gastrointestinal symptoms. Videos with poor bowel preparation were excluded. Personally identifiable information was anonymized prior to analysis.

The dataset included 117 colonoscopy videos totaling 30 h, 57 min, and 46 s, averaging approximately 15 min and 53 s per patient. Overall, the dataset comprised 3,348,994 frames, averaging 28,624 frames per video.

### 2.2. Public Datasets

Representative public datasets used for colonoscopy research include ETIS-Larib [8], CVC-ClinicDB [9], KVASIR-SEG [10], LDPolypVideo [11], KUMC [12], and PolypGen [13]. Each dataset offered unique features and contributions to polyp detection and segmentation tasks: (a) ETIS-Larib Polyp DB contains 196 image frames with binary masks of polyps. (b) CVC-ClinicDB provides 612 images with binary masks extracted from 29 colonoscopy videos. (c) Kvasir-SEG offers 1000 polyp images with corresponding masks. (d) LDPolypVideo includes 160 labeled and 42 unlabeled videos, containing 33,884 images annotated with bounding boxes. (e) KUMC provides 37,899 frames with bounding box annotations. (f) PolypGen contains 1537 polyp images, each accompanied by binary masks.

### 2.3. Data Processing

Colonoscopy videos were used to develop two deep-learning-based AI models. The collected colonoscopy videos (n = 117) from Gangnam Severance Hospital were randomly allocated into three groups: training (57 videos), validation (15 videos), and test sets (45 videos), using a computer-generated randomization procedure to minimize selection bias. The extracted frames from these videos, combined with publicly available datasets (ETIS-Larib, CVC-ClinicDB, KVASIR-SEG, LDPolypVideo, KUMC, PolypGen), were utilized to construct comprehensive image datasets for model training, validation, and testing. The detailed distribution of images across the datasets is summarized in Table 1.

Two deep-learning AI models were developed to compare the effects of integrating real-world clinical data. Model 1 was trained exclusively using publicly available datasets, while Model 2 incorporated additional images obtained from colonoscopy videos collected at Gangnam Severance Hospital. Both models were trained using images containing polyps and normal colonoscopic images without polyps, as outlined in Table 2. The final evaluation was performed using the best-performing model, as determined by accuracy on the validation set.

### 2.4. Semi-Automatic Image Selection

The proposed development process is illustrated in Figure 1. Initially, Model 1 was trained and validated exclusively using public datasets. Subsequently, Model 1 was used to infer frames from 72 colonoscopy videos (57 training, 15 validation) collected from Gangnam Severance Hospital. In this inference step, clinical timestamps indicating the actual removal of polyps during colonoscopy were indirectly used to guide the AI model in identifying frames likely containing polyps. The inference results were automatically categorized into subfolders as frames containing polyps and frames without polyps. Each detected polyp area was highlighted with bounding boxes on corresponding video frames and saved separately, facilitating efficient review by researchers. Frames incorrectly classified by Model 1—such as missed polyps (false negatives) or incorrectly identified polyps (false positives)—were also selectively included to enhance subsequent model training. This combined inference and data-selection approach, termed the semi-automatic image selection process, substantially reduced manual annotation effort, enabling rapid construction of the training dataset for Model 2.

Figure 2 shows representative examples of inference results from Gangnam Severance Hospital using Model 1 trained on public datasets. Each row represents a sequence of consecutive frames. Frames with blue borders indicate instances where polyps were inaccurately detected or completely missed. These misdetected frames were selectively incorporated into the training dataset to enhance the subsequent learning of Model 2.

### 2.5. Algorithm Training, Validation, and Test Sets

A YOLO-based object detection model, a deep learning approach, was employed to detect polyps in the colonoscopy images. Previous studies have shown that various backbones based on convolutional neural networks (CNNs) exhibit similar performance when applied to endoscopic images [2,3,6,12]. Many existing AI models used for colonoscopy have initially been developed for general image recognition tasks and subsequently adapted for specific object detection tasks, including polyp detection.

In this study, we adopted a CNN-based polyp detection model utilizing the Visual Geometry Group (VGG)16 backbone. The VGG16 backbone has been widely applied in medical image analysis and has demonstrated reliable performance in extracting relevant image features in various endoscopic imaging tasks [14,15].

To train Model 1, we augmented the training data (public datasets only) by a factor of 10 using various data augmentation methods, including vertical and horizontal flips, left–right symmetry, random cropping, and rotation. Input image size was standardized at 224 × 224 pixels, and polyp masks and bounding box annotations were consistently preserved during augmentation. Model 1 was trained for 50 epochs with a VGG16 backbone followed by fully connected layers, optimized using the Adam optimizer with an initial learning rate of 1 × 10^−5^. Cross-entropy and activation errors served as error functions for backpropagation.

Model 2 integrated both public datasets and additional images derived from colonoscopy videos obtained at Gangnam Severance Hospital. Frames from these videos were efficiently annotated using a semi-automatic selection process based on preliminary inference by Model 1. Model 2 underwent similar training procedures as Model 1, with validation performance assessed at each epoch. The optimal performing model on the validation set was selected for subsequent testing.

The test dataset comprised 4373 previously unseen images, including 1200 polyp-containing and 3173 non-polyp-containing images, extracted from 45 separate test videos. Specifically, 120 polyps were identified in these test videos, and approximately 10 sequential frames per polyp (at 5-frame intervals, ~1.6 s duration per polyp) were extracted for performance evaluation.

### 2.6. Evaluations

Several metrics were used to evaluate the model performance [16,17]. True positives (TP) represent the number of polyps that were accurately detected by the model. The true negative (TN) indicates the number of normal images correctly classified as normal. A false negative (FN) refers to instances in which a polyp was present but not detected by the model, whereas a false positive (FP) denotes cases in which a polyp was incorrectly identified in a normal image.

Sensitivity (or recall) reflects the proportion of actual polyps that were correctly identified by the model and was computed as TP divided by the sum of TP and FN. Specificity represents the proportion of actual normal images that were correctly classified and was calculated as TN divided by the sum of TN and FP. Accuracy measures the overall proportion of correctly classified images, both polyps and normal, and was determined by dividing the sum of TP and TN by the total number of images (TP + TN + FP + FN). Additionally, the F1 Score, which is the harmonic means of sensitivity and positive predictive value (PPV), provides a balanced measure of precision and recall. The F1 Score was calculated as twice the product of sensitivity and PPV divided by the sum of sensitivity and PPV, where PPV was calculated as TP divided by the sum of TP and FP.



$Sensitivity=TP/\left(TP+FN\right)$



Specificity=TN/(TN+FP)



Accuracy=(TP+TN)/(TP+TN+FP+FN)



$PPV=TP/\left(TP+FP\right)$



F1 score = 2 × (Sensitivity × PPV)/(Sensitivity + PPV)



## 3. Results

The performance of the proposed method was evaluated using the model from the epoch that demonstrated the highest sensitivity and specificity of the validation data. This evaluation was performed over 50 epochs for Model 1, which was trained solely with public datasets, and Model 2, which was trained with both public and additional collected data. Model 1 achieved optimal performance at epoch 37 based on validation accuracy, whereas Model 2’s best performance occurred at epoch 35. Table 3 presents the polyp detection performance on the test data using the model saved at epoch 37 for Model 1 and the models saved at 5-epoch intervals from epoch 5 to epoch 45 for Model 2. Figure 3 demonstrates stable learning curves for both models without signs of overfitting, indicating effective generalization throughout the training process. Specifically, across 50 epochs, Model 1 showed a mean validation accuracy of 0.943 (standard deviation [SD] = 0.013), and Model 2 showed a mean validation accuracy of 0.939 (SD = 0.012). Since Model 1 exhibited its highest validation accuracy at epoch 37, this epoch was selected to compare performance against Model 2, trained with integrated data. According to Table 3, Model 2 consistently outperformed Model 1 across most evaluated metrics at the selected epochs, confirming the benefits of incorporating additional colonoscopy video data into the training process.

The test results revealed that Model 1, trained only with public datasets, had a sensitivity of 0.778 for detecting polyps, indicating reasonable performance. However, it exhibited a low specificity of 0.059 due to significant false detections, which were attributed to differences in image quality between the collected data and public datasets. In contrast, Model 2, which incorporated additional collected data into the training process, demonstrated an improved performance by addressing these environmental differences. Specifically, Model 2 achieved a sensitivity of 0.906, a specificity of 0.960, and an accuracy of 0.945 at epoch 35, reflecting a notable enhancement in detection capability.

In real-time colonoscopy applications, the polyp detection model must minimize false detections in normal images and ensure that all polyps are accurately identified to prevent them from being overlooked. Frequent false detections can lead to examiner fatigue and reduce the reliability of the detection model. Conversely, accurate detection of polyps occasionally provides opportunities to identify and remove lesions that might otherwise be missed. Table 4 details the number and detection rates of polyps that were never detected out of the 120 polyps, each represented by 10 images, amounting to 1200 test images. Figure 4 shows examples of the 120 undetected polyps. Figure 5 shows the number of polyps accurately detected in each epoch. The results indicated that Model 2, which utilized additional collected data, consistently demonstrated higher sensitivity than Model 1, which relied solely on public datasets. This suggests that Model 2 offers an improved continuity of polyp detection throughout the video sequences.

## 4. Discussion

Recent advances in AI-based CADe and CADx have significantly improved the detection and characterization of colorectal polyps during colonoscopy, potentially enhancing clinical outcomes through earlier and more accurate diagnosis [18,19]. However, AI models trained exclusively on publicly available datasets frequently demonstrate limited specificity when applied to real-world clinical scenarios, mainly due to differences in image quality, diversity, and endoscopic equipment [20,21].

In this study, we proposed a simple and efficient approach to enhance the performance of AI models by integrating real-world colonoscopy video data into the training process. A distinctive feature of our method was the semi-automatic annotation technique, which utilized preliminary inference results from a publicly trained AI model to rapidly and automatically categorize colonoscopy video frames into polyp-containing and non-polyp-containing categories. Although semi-automatic annotation tools, such as CVAT, are widely used in computer vision, our approach specifically employed AI-guided inference to minimize manual intervention by expert endoscopists, thereby significantly reducing the traditionally labor-intensive annotation workload [22].

By integrating real patient colonoscopy data through this approach, we observed notable improvements in both sensitivity and specificity compared to a model trained solely on public datasets. This highlights that our method not only improves AI model performance but also offers practical advantages by substantially reducing manual annotation efforts.

However, several limitations of our study warrant careful consideration. First, this study used colonoscopy videos from a single institution employing a standardized endoscopic imaging system (Olympus Evia Exera III CV-190, Olympus Corporation, Tokyo, Japan). Therefore, generalizability of our results to other clinical environments and various endoscopic equipment has not yet been confirmed [23]. Future studies should include external validation using multi-center datasets to verify broader clinical applicability and reproducibility of results. Additionally, the study lacked detailed patient characteristics data, including age, gender, clinical indications, polyp morphology, size, location, and histopathological results. Consequently, we could not evaluate the AI model’s performance across different lesion types or diverse patient subpopulations, potentially limiting the clinical interpretation of our results. Further research incorporating comprehensive clinical and histopathological data is required to better understand the AI model’s performance across various lesion types and patient subpopulations, which will be crucial for identifying potential areas of improvement and enhancing clinical relevance. Furthermore, our evaluation focused primarily on per-frame metrics such as sensitivity and specificity, which may not fully reflect clinical relevance. To better evaluate AI systems in clinical settings, future studies should incorporate lesion-level sensitivity, precision metrics per detected polyp, and precision-recall curves or receiver operating characteristic (ROC) analyses.

## 5. Conclusions

In conclusion, our study introduces a practically meaningful, efficient, and scalable approach to significantly enhance AI model performance for colonoscopy polyp detection using real-world colonoscopy video data. The semi-automatic annotation method employed could substantially reduce the traditionally labor-intensive manual annotation process typically performed by expert endoscopists, making it an attractive strategy for practical clinical implementation. However, further external validation, detailed patient and lesion characteristic analyses, and lesion-level performance assessments are necessary to confirm and maximize the clinical utility and generalizability of our approach.

## Figures and Tables

**Figure 1 diagnostics-15-00901-f001:**
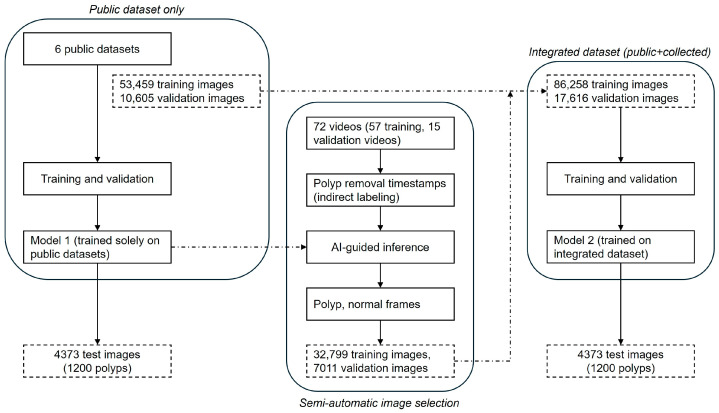
Process of developing the initial model with public datasets and constructing additional datasets.

**Figure 2 diagnostics-15-00901-f002:**
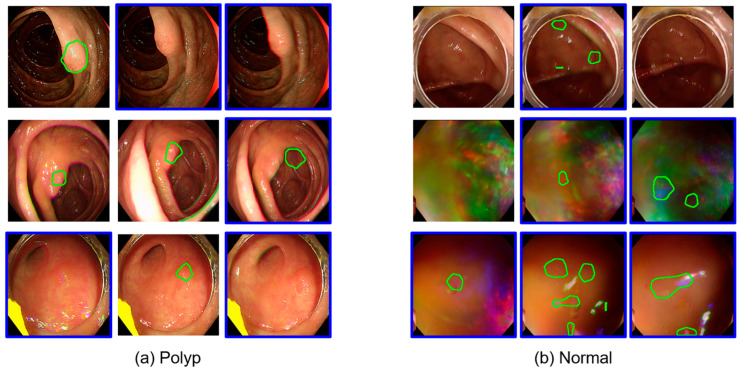
Examples of resulting images from inference in Model 1. The images include detected polyps (**a**) and normal tissue (**b**). Blue squares indicate data selected for further training.

**Figure 3 diagnostics-15-00901-f003:**
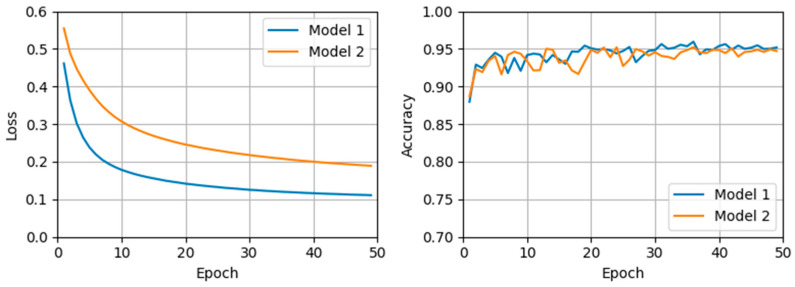
Training loss and validation accuracy per epoch for Model 1 and Model 2.

**Figure 4 diagnostics-15-00901-f004:**
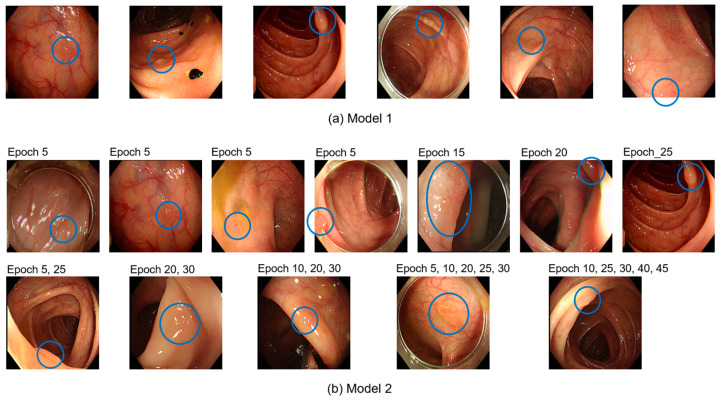
Example of polyps (blue circle) that were not detected by either Model 1 (**a**) or Model 2 (**b**) among 120 test polyps.

**Figure 5 diagnostics-15-00901-f005:**
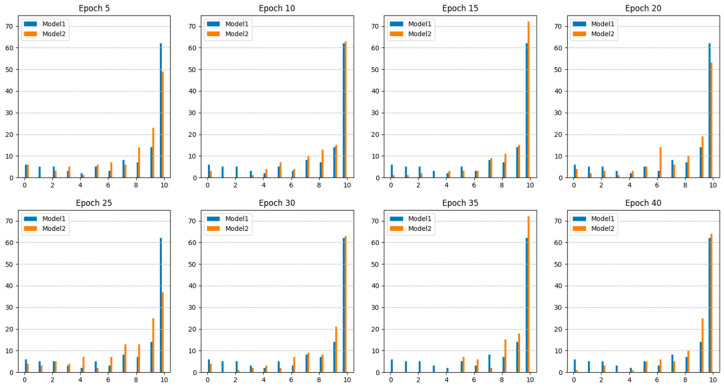
The number of TPs for polyps in each epoch. The *X*-axis represents the number of TPs, while the *Y*-axis indicates the number of polyps.

**Table 1 diagnostics-15-00901-t001:** Summary of the number of images used for training, validation, and test.

Dataset	Training (polyps)	Validation (polyps)	Test (polyps)	Total (polyps)
A. ETIS-Larib	196 (196)	-	-	196 (196)
B. CVC-ClinicDB	612 (612)	-	-	612 (612)
C. KVASIR-SEG	900 (900)	-	-	900 (900)
D. LDPolypVideo	23,723 (1604)	6191 (462)	-	29,914 (2066)
E. KUMC	27,048 (27,048)	4214 (4214)	-	31,262 (31,262)
F. PolypGen	980 (980)	200 (200)	-	1180 (1180)
G. Gangnam Severance Hospital	32,799 (9353)	7011 (623)	4373 (1200)	44,183 (11,176)
Total	86,258 (40,693)	17,616 (5499)	4373 (1200)	108,247 (47,392)

**Table 2 diagnostics-15-00901-t002:** Dataset configuration used to develop AI models.

Model	Dataset	Number of Images (polyps)			
		Training	Validation	Test	Total
1	A. + B + C + D + E + F (Public dataset only)	53,459 (31,340)	10,605 (4876)	4373 (1200)	68,437 (37,416)
2	A. + B + C + D + E + F + G (Public + Collected dataset)	86,258 (40,693)	17,616 (5499)	4373 (1200)	108,247 (47,392)

**Table 3 diagnostics-15-00901-t003:** Comparison of polyp detection performance between the model trained only with public datasets and the integrated model that includes additional data from colonoscopy videos at Gangnam Severance Hospital.

	Model 1	Model 2								
	Epoch 37	Epoch 5	Epoch 10	Epoch 15	Epoch 20	Epoch 25	Epoch 30	Epoch 35	Epoch 40	Epoch 45
TP	933	948	1017	1056	955	895	1018	1087	1051	1042
FN	267	252	183	144	245	305	182	113	149	158
FP	3647	537	365	427	308	160	184	132	180	148
TN	230	3104	3100	3100	3141	3155	3149	3142	3144	3144
Sensitivity	0.778	0.790	0.848	0.880	0.796	0.746	0.848	0.906	0.876	0.868
Specificity	0.059	0.853	0.895	0.879	0.911	0.952	0.945	0.960	0.946	0.955
PPV	0.204	0.638	0.736	0.712	0.756	0.848	0.847	0.892	0.854	0.876
F1 score	0.323	0.706	0.788	0.787	0.775	0.794	0.848	0.899	0.865	0.872
Accuracy	0.229	0.837	0.883	0.879	0.881	0.897	0.919	0.945	0.927	0.932

**Table 4 diagnostics-15-00901-t004:** Comparison of polyp detection performance between Model 1 and Model 2.

	Model 1	Model 2								
	Epoch 37	Epoch 5	Epoch 10	Epoch 15	Epoch 20	Epoch 25	Epoch 30	Epoch 35	Epoch 40	Epoch 45
Undetected polyp	6	6	3	1	4	4	4	0	1	1
Detection rate (%)	95.00	95.00	97.50	99.17	96.67	96.67	96.67	100.0	99.17	99.17

## Data Availability

The original contributions presented in this study are included in the article. Further inquiries can be directed to the corresponding authors.

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
