# Peer review of "Real-World Colonoscopy Video Integration to Improve Artificial Intelligence Polyp Detection Performance and Reduce Manual Annotation Labor"

_diagnostics, 2025, doi:10.3390/diagnostics15070901_

Round 1
Reviewer 1 Report
Comments and Suggestions for Authors
In this study, the authors proposed a new a semi-automatic annotation method using real colonoscopy videos to enhance AI model performance and reduce manual labeling labor. However, this work is merit, some minor recommendation for Improvement should be addressed.
- Replace figure 1 that shows the architetcure of the model earlier to be in page 3 instead of page 5
- In line 139, the author sys he used VGG model but ,There are many variation of VGG models, the author shoud indicate which one is used VGG16 or VGG19
- Its preferable to map section 2.6 to equations for the performance metrcis used in evaluation.
- In line 184 , the authos sya that model 1 achived optimal performance at epoch 37, the author should clarify what is the metric that he relay on to comape the perrformance, also by examining the table 3 and table 4 it is not show that epoch 37 is better than others. I think the author should double check this value or give more explansion to link the discussion section with the results exists in theses tables.
- Its bettere to move Table 1 and table 2 which give detailed explansion of datasets to be under section 2.2 or section 2.3
- Its better to add a separet section for conclusion instead of merging it with discussion section
- The training and testing for loss curve should be added to show is the propsed model overfitted or not.
Author Response
Comment 1: Replace figure 1 that shows the architecture of the model earlier to be in page 3 instead of page 5
Response 1: We have relocated Figure 1 (model architecture) from page 5 to page 3 for clearer presentation earlier in the manuscript.
Comment 2: In line 139, the author says he used VGG model but, There are many variations of VGG models, the author should indicate which one is used VGG16 or VGG19
Response 2: We clarified the specific VGG16 backbone used and revised the manuscript.
Comment 3: Its preferable to map section 2.6 to equations for the performance metrics used in evaluation.
Response 3: As recommended, we included explicit equations for clarity.
Comment 4: In line 184, the authors say that model 1 achieved optimal performance at epoch 37, the author should clarify what is the metric that he relays on to compare the performance, also by examining table 3 and table 4 it is not show that epoch 37 is better than others. I think the author should double check this value or give more explanation to link the discussion section with the results in these tables.
Response 4: We rechecked and clarified our selection criteria. Additional clarification and justification have been included in the Results sections.
Comment 5: It’s better to move Table 1 and table 2 which give detailed explanation of datasets to be under section 2.2 or section 2.3
Response 5: We repositioned Tables 1 and 2 following section 2.3 (Data processing) to enhance readability and coherence.
Comment 6: It’s better to add a separate section for conclusion instead of merging it with discussion section
Response 6: A dedicated "Conclusion" section has been added at the end of the manuscript, succinctly summarizing key findings, clinical implications, and future research directions.
Comment 7: The training and testing for loss curve should be added to show is the proposed model overfitted or not.
Response 7: We have added a new figure illustrating the training loss and validation accuracy curves over the epochs, clearly demonstrating that our proposed model does not exhibit overfitting. A detailed explanation interpreting these loss curves has also been provided within the result section.
Reviewer 2 Report
Comments and Suggestions for Authors
Based on the content provided from the research article, the following are the weak points of this study:
- The colonoscopy videos used in this study were collected from a single institution.
- The study did not collect comprehensive patient data such as age, gender, clinical indications, polyp morphology, size, location, and histopathological results.
- The evaluation primarily focused on **per-frame metrics** such as sensitivity and specificity.
- The study relied on internal validation using data from the same institution and did not perform external validation on independent datasets.
Author Response
Comment 1: The colonoscopy videos used in this study were collected from a single institution.
Response 1: We acknowledge this limitation and explicitly mentioned it in our Discussion section. While our findings demonstrate significant improvements in model performance, external validation using multicenter datasets remains necessary to confirm the generalizability of our results. We have clearly noted this as a recommendation for future research.
Comment 2: The study did not collect comprehensive patient data such as age, gender, clinical indications, polyp morphology, size, location, and histopathological results.
Response 2: We recognize that the absence of detailed patient and lesion characteristics is a limitation, potentially restricting the interpretability and clinical applicability of our findings. We have acknowledged this in the Discussion section, highlighting the need for future research that integrates comprehensive clinical data to better evaluate the model's performance across diverse patient subpopulations and lesion characteristics.
Comment 3: The evaluation primarily focused on **per-frame metrics** such as sensitivity and specificity.
Response 3: We appreciate this valuable feedback. While our initial analysis emphasized per-frame metrics, we agree that lesion-level evaluations would provide further insights into clinical utility. Future studies are planned to include these comprehensive evaluations. We have expanded the Discussion section accordingly to clarify these points and highlight future directions to address these limitations.
Comment 4: The study relied on internal validation using data from the same institution and did not perform external validation on independent datasets.
Response 4: We acknowledge this limitation and explicitly mentioned it in our Discussion section. While our findings demonstrate significant improvements in model performance, external validation using multicenter datasets remains necessary to confirm the generalizability of our results. We have clearly noted this as a recommendation for future research.